# Algorithms and Hardness for Active Learning on Graphs

**Vincent Cohen-Addad** [1] **Silvio Lattanzi** [1] **Simon Meierhans** [2]

## Abstract

We study the offline active learning problem on graphs. In this problem, one seeks to select $k$ vertices whose labels are best suited for predicting the labels of all the other vertices in the graph. Guillory and Bilmes (Guillory & Bilmes, 2009) introduced a natural theoretical model motivated by a label smoothness assumption. Prior to our work, algorithms with theoretical guarantees were only known for restricted graph types such as trees (Cesa-Bianchi et al., 2010) despite the models simplicity. We present the first $O(\log n)$-resource augmented algorithm for general weighted graphs. To complement our algorithm, we show constant hardness of approximation.

## 1. Introduction

Graph learning has a wide array of applications across various domains spanning Web Spam detection, genomics, text classification, face detection and many more (Chang & Yeung, 2006; Goldberg & Zhu, 2006; Herbster & Lever, 2009; Shin et al., 2009). In active learning, one is presented with a large unlabeled data set and given the opportunity to uncover the labels of some small number of points. Obtaining labeled data points is often costly, and therefore these should be chosen carefully. Guillory and Bilmes (Guillory & Bilmes, 2009) introduced a simple graph based abstraction where every vertex corresponds to a data point, and edges encode similarities between data points. For vertex labels $\boldsymbol{y} \in \{0,1\}^V$, they assume that

$$\sum_{(i,j)\in E} \boldsymbol{w}(i,j) \cdot |\boldsymbol{y}(i) - \boldsymbol{y}(j)|$$

is small and show that then in order to optimize label selection one can then focus on selecting a set of nodes $L$ for the objective

$$\Psi(L) \stackrel{\text{def}}{=} \min_{C \subseteq V \setminus L} \boldsymbol{w}(C, V \setminus C)/|C|$$

[1]Google [2]ETH Zurich, Switzerland. Correspondence to: Simon Meierhans <mesimon@inf.ethz.ch>.

*Proceedings of the $42^{nd}$ International Conference on Machine Learning*, Vancouver, Canada. PMLR 267, 2025. Copyright 2025 by the author(s).

on the weighted similarity graph $G = (V, E, \boldsymbol{w})$ where the goal is to maximize $\Psi(\cdot)$ subject to a cardinality constraint $k$

$$\max_{L \subset V : |L| \leq k} \Psi(L).$$

Concretley, they show that for any consistent labeling $\hat{\boldsymbol{y}}$, i.e. $\boldsymbol{y}(v) = \hat{\boldsymbol{y}}(v)$ for all $v \in L$, we have

$$\|\boldsymbol{y} - \hat{\boldsymbol{y}}\|^2 \leq \frac{1}{2\Psi(L)}\bigg( \sum_{(i,j)\in E} \boldsymbol{w}(i,j) \cdot |\boldsymbol{y}(i) - \boldsymbol{y}(j)| \\ + \sum_{(i,j)\in E} \boldsymbol{w}(i,j) \cdot |\hat{\boldsymbol{y}}(i) - \hat{\boldsymbol{y}}(j)| \bigg)$$

where the first sum is small by assumption and the second sum is (approximately) minimized by a suitable prediction algorithm (See Theorem 1 in (Guillory & Bilmes, 2009)). We therefore focus on maximizing $\Psi(L)$ in this article.

Intuitively, the objective asks to position the labeled data points such that disconnecting a large chunk of unlabeled data points cuts a lot of edge weight. (Guillory & Bilmes, 2009) present some evidence that solving this problem exactly might be hard, and give practical heuristic algorithms.

Cesa-Bianchi, Gentile, Vitale and Zappella presented the first algorithm with theoretical guarantees under the assumption that $G$ is an unweighted tree (Cesa-Bianchi et al., 2010). Their algorithm introduces constant error. In the conclusion, they speculate:

> "We also believe that an extension to general graphs of our algorithm does actually exist. However, the complexity of the methods employed in (Guillory & Bilmes, 2009) suggests that techniques based on minimizing $\Psi(L)$ on general graphs are computationally very expensive."
> (Cesa-Bianchi et al., 2010), page 17.

In the meantime, they suggest sampling a random spanning tree as a plausible heuristic for applying their algorithm to general unweighted graphs.

In this article, we show that an efficient and reasonably simple resource augmented algorithm for maximizing $\Psi(L)$ on a general graphs exists.

**Theorem 1.1.** *Given a graph $G = (V, E, \boldsymbol{w})$ with polynomially bounded[1] integral edge weights and a budget $k$, there exists an algorithm that returns a set $L'$ such that*

- $|L'| \leq O(\log |V|) \cdot k$ *and*

- $\Psi(L') \geq \max_{L \subset V : |L| \leq k} \Psi(L).$

*The algorithm runs in time $k \cdot |V| \cdot (|V| + |E|)^{1+o(1)}$ where $o(1) \to 0$ as $|V| \to \infty$.*

We point out that such resource augmented algorithms are loosely competitive, i.e. our algorithm approximates $\max_{|L| < k} \Psi(k)$ well for most budgets (Young, 2002; Roughgarden, 2021).

To complement our algorithm, we also give the first hardness result[2] for maximizing $\Psi(L)$.

**Theorem 1.2** (Hardness of Approximation)**.** *It is NP-hard to decide if*

- $\max_{L \subset V : |L| \leq k} \Psi(L) \leq 2$ *or*

- $\max_{L \subset V : |L| \leq k} \Psi(L) \geq 3$

*for general unweighted graphs and labelling budgets $k$.*

We complement our theoretical results with proof-of-concept experiments on both synthetic and real world graphs showing that our algorithm has better experimental performances than the heuristics presented in (Guillory & Bilmes, 2009; Cesa-Bianchi et al., 2010).

**Technical overview.** The most direct template algorithm for approximating $\max_{|L| \leq k} \Psi(L)$ is to build the set $L$ one by one. Our algorithm and the previous algorithms of (Guillory & Bilmes, 2009) and (Cesa-Bianchi et al., 2010) all follow this approach. Because the $\Psi(\cdot)$ objective is neither submodular nor supermodular[3], it is a priori unclear how to choose the next vertex to add without resorting to heuristics.[4]

To resolve this issue, we develop a new perspective motivated by the densest subgraph problem. In this well-studied problem, one seeks to find an induced subgraph with maximum average degree (Goldberg, 1984; Boob et al., 2020; Chekuri et al., 2022). Given some target degree $\delta$, (Goldberg, 1984) proposed a flow gadget for deciding if a subgraph of average degree $\delta$ exists. Intuitively, a dense subgraph is a cluster which has few connections to the rest of

---

[1]In the number of vertices $|V|$.

[2]To the best of our knowledge.

[3]See Appendix A.

[4]E.g. (Guillory & Bilmes, 2009) suggest adding a random vertex from a current witness cut $C$ minimizing $\min_{C \subseteq V \setminus L} \boldsymbol{w}(C, V \setminus C)/|C|$.

the graph compared to its size. This reveals some similarity with our problem. Loosely speaking, we aim to select the set $L$ such that $G[V \setminus L]$ does not contain a very dense subgraph. This motivates us to construct a similar flow gadget for maximizing $\Psi(\cdot)$.

In our flow gadget, adding vertices to $L$ corresponds to adding new edges, which increases the maximum flow. This yields a simple greedy rule we exploit in our algorithm: Add the vertex whose edge increases the maximum flow the most to $L$. This measure is submodular, and we are thus able to give guarantees via standard submodular function maximization.

Finally, our flow gadget still requires a guess for the value $\max_{|L| \leq k} \Psi(L)$ similar to the guessed average degree $\delta$ for densest subgraph. We use binary search to obtain our full algorithm.

**Related work.** We restrict our attention to graph problems, and point the reader to (Settles, 2009; Ren et al., 2021) for surveys on classical and deep active learning.

Multiple classic approaches for the passive problem where $L$ is fixed have been proposed (Blum & Chawla, 2001; Blum et al., 2004; Belkin et al., 2004; Bengio et al., 2006). These introduce the use of smoothness assumptions on the labels.

The active case where the set $L$ is not fixed has received considerable attention (Zhu et al., 2003; Guillory & Bilmes, 2009; Cesa-Bianchi et al., 2010; Dasarathy et al., 2015) in the case where the vertices are either selected one-by-one or all at once. Recent approaches have incorporated deep learning (Mac Aodha et al., 2014; Kushnir & Venturi, 2020; Hu et al., 2020; Zhang et al., 2022b;a). In (Zhang et al., 2022a), learned functions are combined with graph based combinatorial techniques based on (Dasarathy et al., 2015), showing that interesting synergies can emerge when classic approaches are combined with deep learning. Unfortunately, no previous work provides any theoretical guarantees on general graphs.

**Roadmap.** In Section 2, we introduce some standard notation. Then, we formally introduce the graph label selection problem in Section 3. In Section 4 we reduce the problem to an equivalent problem involving flows, and in Section 5 we present a greedy approximation algorithm exploiting that flow perspective. We show that the problem is hard to approximate in Section 6, and present experiments on small graphs in Section 7. Finally, in Section 8 we show that our algorithm can be generalized to vertex importances.

## 2. Preliminaries

**Graphs.** We consider undirected weighted graphs $G = (V, E, \boldsymbol{w})$ where $V$ denotes the vertex set and $E \subseteq V \times V$

denotes the edge set. The vector $\boldsymbol{w} \in \mathbb{R}^E_{\geq 0}$ contains integral edge weights.

We associate an edge-vertex incidence matrix $\boldsymbol{B} \in \mathbb{R}^{E \times V}$ with the graph $G$. This matrix has exactly two nonzero entries per row. For the edge $e = (u,v)$, we have $B(e,u) = -1$ and $\boldsymbol{B}(e,v) = 1$. Because the edges of $G$ are undirected, we associate some arbitrary direction with the edges to define the matrix $\boldsymbol{B}$.

**Induced subgraphs.** Given some set $A \subset V$, we let $G[A]$ denote the induced subgraph on $A$, i.e. the graph obtained when restricting to the set of vertices in $A$.

**Cuts.** For a set $C \subset V$ sometimes referred to as a cut, we let $E(C, V \setminus C)$ denote the subset of $E$ that only contains edges with exactly one endpoint in $C$, i.e. the set of edges that cross the cut. Furthermore, we let $\boldsymbol{w}(C, V \setminus C) \stackrel{\text{def}}{=} \sum_{e \in E(C, V \setminus C)} \boldsymbol{w}(e)$ denote the sum of the weights of the edges going across the cut.

**Flows.** For a graph $G = (V, E, \boldsymbol{w})$ with edge vertex incidence matrix $\boldsymbol{B}$, we call a flow $\boldsymbol{f} \in \mathbb{R}^E$ feasible if $-\boldsymbol{w} \leq \boldsymbol{f} \leq \boldsymbol{w}$. We say such a flow routes the demand $\boldsymbol{d} = \boldsymbol{B}^\top \boldsymbol{f}$. In this article, we are interested in flows that route s-t demands, i.e. demand vectors $\boldsymbol{d} = \alpha(\mathbf{1}_t - \mathbf{1}_s)$. Maximizing the amount of flow $\alpha > 0$ such that a feasible flow routing $\boldsymbol{d}$ exists is called the maximum flow problem. Due to recent progress on the more general min-cost flow problem (Chen et al., 2023; van den Brand et al., 2023; Chen et al., 2024; van den Brand et al., 2024), this problem can be solved in time $m \cdot e^{O(\log^{3/4} m \log \log m)} \log U$ where $U$ is an upper bound on the edge weights of $G$ (van den Brand et al., 2024).

Every s-t flow can be decomposed into a set of s-t paths and cycles, and removing all cycles does not change feasibility and the routed demands. By duality, the maximum amount of flow that can be sent from $s$ to $t$ with a feasible flow is equal to the min-cut $\min_{C \in V \setminus \{t\} : s \in C} \boldsymbol{w}(C, V \setminus C)$.

# 3. The Graph Label Selection Problem

In this section, we formally define the graph label selection problem. We first define the sparsity of a cut.

**Definition 3.1** (Sparsity)**.** For a graph $G = (V, E, \boldsymbol{w})$ and a cut $\emptyset \subset C \subset V$, we let

$$\Psi_C \stackrel{\text{def}}{=} \frac{\boldsymbol{w}(C, V \setminus C)}{|C|}$$

Furthermore, we let $\Psi_\emptyset \stackrel{\text{def}}{=} \infty$, and $\Psi_V \stackrel{\text{def}}{=} 0$.

Notice that in Definition 3.1 the cardinality of $C$ is not bounded by $|V|/2$ in contrast to the usual definition of sparsity in the context of vertex expansion. We let the empty

cut and the cut containing all vertices have sparsity $\infty$ and $0$ respectively for notational convenience.

We then define the graph label selection objective as introduced in (Guillory & Bilmes, 2009), and introduce a slack parameter on the cardinality of the set of selected vertices similar to (Cesa-Bianchi et al., 2010).

**Definition 3.2** (Graph Label Selection (GLS))**.** Given a graph $G = (V, E, \boldsymbol{w})$ and a budget $k \leq |V|$, we let

$$\Psi(L) \stackrel{\text{def}}{=} \min_{C \subseteq V \setminus L} \Psi_C$$

denote the minimum sparsity $\Psi_C^G$ obtained by a cut $C$ that does not include any vertex in $L$. The learning set selection objective is then given by

$$\max_{L \subseteq V : |L| \leq k} \Psi(L).$$

We call any solution $L'$ such that $|L'| \leq s \cdot k$ and

$$(1 + \epsilon)\Psi(L') \geq \max_{L \subseteq V : |L| \leq k} \Psi(L)$$

a solution with slack $s \in \mathbb{R}_{\geq 0}$ and error $\epsilon$.

We then define the associated decision problem.

**Definition 3.3** (Thresholded Graph Label Selection (T-GLS))**.** Given a graph $G = (V, E, \boldsymbol{w})$, a budget $k \leq |V|$ and a threshold $\tau \in \mathbb{R}_{>0}$, the thresholded graph label selection problem with slack $s$ is to either

- certify that $\max_{L \subseteq V : |L| \leq k} \Psi(L) < \tau$ or

- output a set $L$ with $|L| \leq s \cdot k$ such that $\Psi(L) \geq \tau$.

**Solving GLS and weight range compression.** Given a (slack $s$) solution to the T-GLS problem, a (slack $s$) solution to the graph label selection problem can be obtained in a straightforward fashion via binary search.

**Lemma 3.4.** *Assume to be given an algorithm* T-GLS$(G, \tau, k)$ *that solves the T-GLS problem with slack $s$. Then, there exists an algorithm that solves the GLS problem for a graph $G$ with slack $s$ using $O(\log nU)$ calls to* T-GLS$(G, \tau, k)$ *where $U$ is an upper bound on the edge weights in $G$.*

*Proof.* The smallest non-trivial threshold is $1/n$ and the largest is $n \cdot U$. Furthermore, all thresholds are representable as fractions $\alpha/\beta$ where $\alpha \in [nU]$ and $\beta \in [n]$. Therefore, a binary search is guaranteed to find a solution $L'$ of slack $s$ after $O(\log nU)$ steps. $\square$

Given Lemma 3.4, we focus on obtaining an algorithm for solving the T-GLS problem.

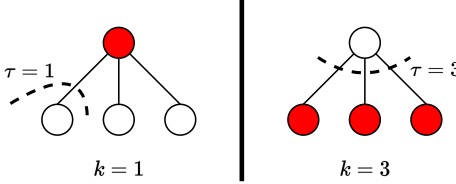

*Figure 1.* On the left hand side, the best vertex to select into $L$ given budget $k = 1$ is labeled in red. On the right hand side, the best vertices to select given a budget of $k = 3$ are labeled in red. For this simple graph, selecting the initially best vertex can turn out to be a mistake. The amount of error can be increased by increasing the size of the star. See also Appendix A.

## 4. Graph Label Selection: A Flow Problem in Disguise

In this section, we take on a flow perspective of the T-GLS problem (Definition 3.3). This yields a new and powerful way to evaluate the usefulness of vertices in conjunction with an existing label set $L$. Although the GLS problem (Definition 3.2) is neither submodular nor supermodular, this new measure of usefulness of vertices with respect to a particular threshold is submodular. We refer the interested reader to Figure 1 for a simple example illustrating some pitfalls and to Appendix A for a more thorough explanation of submodularity.

Our approach is inspired by Goldbergs reduction from densest subgraph to maximum flow (Goldberg, 1984). The densest subgraph problem asks to find a set $C \subset V$ such that $\frac{\sum_{e \in E_{G[C]}} \boldsymbol{w}(e)}{|C|}$ is maximized, which corresponds to the induced subgraph with largest average degree. If the graph is $\Delta$-regular, i.e. $\deg_G(v) = \Delta$, then the density of the subgraph $G[C]$ is directly related to the sparsity of the cut $C$

$$\max_{C \subset V} \underbrace{\frac{\sum_{e \in E_{G[C]}} \boldsymbol{w}(e)}{|C|}}_{\text{density}} = \Delta - \min_{C \subset V} \underbrace{\frac{\boldsymbol{w}(C, V \setminus C)}{|C|}}_{\text{sparsity}}.$$

We next present a flow gadget which captures the sparsity of cuts directly (See also Figure 2).

**Definition 4.1** (Flow Graph). Assume to be given an undirected graph $G = (V, E, \boldsymbol{w})$, a set $L$ and a threshold $\tau$. Let $W$ denote the sum of the edge weights of $G$. We then construct a graph $\widehat{G}_{L,\tau}$ as follows.

- (Vertex set). The vertex set of $\widehat{G}_{L,\tau}$ is $V \cup \{s, t\}$.

- (Copy). Every edge $e \in E$ is present in $\widehat{G}_{L,\tau}$ with the same weight.

- (Source). For every vertex $v \in V$, the graph $\widehat{G}_{L,\tau}$ contains an edge $(s, v)$ with weight $\tau$.

- (Sink). For every vertex $l \in L$, the graph $\widehat{G}_{L,\tau}$ contains an edge $(l, t)$ with weight $W + \tau + 1$.

**Observation 4.2.** *We observe that $\widehat{G}_{L',\tau}$ is a subgraph of $\widehat{G}_{L,\tau}$ if $L' \subseteq L$.*

We recall the definition of min-cuts from the preliminaries.

**Definition 4.3** (*s-t* min-cut). Given a graph $G = (V, E, \boldsymbol{w})$ and two vertices $s, t$, the *s-t* min-cut is given by

$$\text{mincut}_G(s, t) \stackrel{\text{def}}{=} \min_{C \subset V \setminus \{t\} : s \in C} \boldsymbol{w}(C, V \setminus C).$$

Next, we relate min-cuts in graphs $\widehat{G}_{L,\tau}$ to the T-GLS problem. We first state a simple observation.

**Observation 4.4.** *$\text{mincut}_{\widehat{G}_{L,\tau}}(s, t) \leq n \cdot \tau$*

*Proof.* The cut $\{s\}$ achieves this value regardless of $L$. $\square$

**Claim 4.5.** *If $\text{mincut}_{\widehat{G}_{L,\tau}}(s, t) < n \cdot \tau$, then $\Psi(L) < \tau$.*

*Proof.* Fix a cut $S \subseteq V$ that achieves the minimum value $\boldsymbol{w}(S, V_{\widehat{G}_{L,\tau}} \setminus S) = \text{mincut}_{\widehat{G}_{L,\tau}}(s, t)$. Then, let $C = S \cap V$.

We first show that $L \cap C = \emptyset$. Assume for the sake of a contradiction that $v \in L \cap C$. Then, the edge $(v, t)$ is in the cut with value $W + \tau + 1$. But removing the vertex $v$ from the cut merely adds edges of weight $\deg_G(v) + \tau$ to the cut while removing an edge of weight $W + \tau + 1$. Since $\deg_G(v) + \tau < W + \tau + 1$, this contradicts the minimality of $S$.

We then have

$$\underbrace{\boldsymbol{w}(C, V \setminus C)}_{\text{graph cut}} + \underbrace{(|V| - |C|) \cdot \tau}_{\text{source cut}} = \text{mincut}_{\widehat{G}_{L,\tau}}(s, t)$$
$$< \tau \cdot |V|$$

by Definition 4.1 and our assumption on the value of the min-cut. We directly obtain

$$\boldsymbol{w}(C, V \setminus C) < \tau \cdot |C|$$

by canceling terms. Thus, $C$ is a sparser than $\tau$ cut that does not contain any vertex in $L$, and $\Psi(L) \leq \boldsymbol{w}(C, V \setminus C)/|C| < \tau$ by Definition 3.2. $\square$

**Claim 4.6.** *If $\Psi(L) < \tau$, then $\text{mincut}_{\widehat{G}_{L,\tau}}(s, t) < n \cdot \tau$.*

*Proof.* Consider a set $C \subseteq V$ that certifies $\Psi(L) < \tau$, i.e. a set $C$ such that $\boldsymbol{w}(C, V \setminus C) < \tau \cdot |C|$. Then the cut $\{s\} \cup C$ has value $n \cdot \tau + \boldsymbol{w}(C, V \setminus C) - \tau \cdot |C| < n \cdot \tau$. $\square$

Together, Claim 4.5 and Claim 4.6 show an equivalence between *s-t* min-cuts in $\widehat{G}_{L,\tau}$ and the T-GLS problem (Definition 3.3).

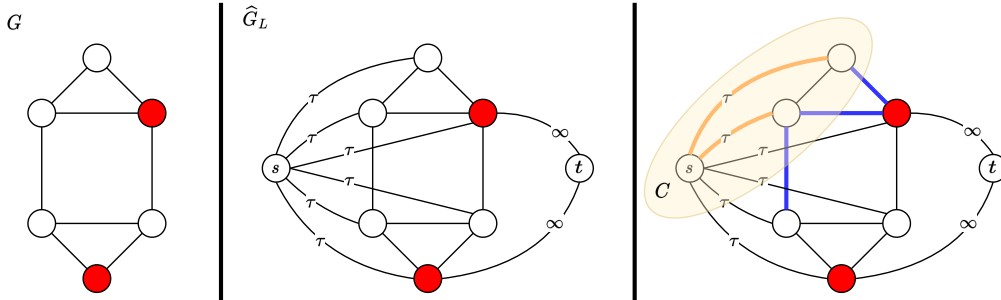

*Figure 2.* On the left, a graph $G$ is displayed with some vertices in $L$ which are displayed in red. To obtain the graph $\widehat{G}_{L,\tau}$ as in Definition 4.1, the vertices $s$ and $t$ are added, the source $s$ is connected to every vertex with an edge of weight $\tau$, and every vertex in $L$ is connected to the sink with an edge of infinite weight $\infty$. Now, consider an $s$-$t$ cut whose left side does not contain any vertices in $L$, such as $C$ in the final panel. We observe that total weight of the blue edges is $\boldsymbol{w}(C \cap V, V \setminus C)$ and the total weight of the orange edges is $|C \cap V| \cdot \tau$. If the weight of the blue edges is less than the weight of the orange edges, then $\Psi(L) < \tau$.

---

**Lemma 4.7.** *There is a set $L$ of size $|L| \leq k$ such that $\Psi(L) \geq \tau$ if and only if there exists a set $L$ with $|L| \leq k$ such that $mincut_{\widehat{G}_{L,\tau}}(s,t) = n \cdot \tau$.*

*Proof.* The lemma directly follows from Claim 4.5 and Claim 4.6. □

By the min-cut max-flow duality, we observe the following corollary.

**Corollary 4.8.** *There is a set $L$ of size $|L| \leq k$ such that $\Psi(L) \geq \tau$ if and only if there exists a set of $L$ with $|L| \leq k$ such that the $s$-$t$ max-flow in $\widehat{G}_{L,\tau}$ is $n \cdot \tau$.*

## 5. Greedy Approximately Solves Label Selection

Given the flow perspective obtained in the previous section, a natural greedy criterion for selecting vertices to add to $L$ presents itself: Simply choose the vertex which increases the maximum-flow the most.

**Greedy algorithm.** In this paragraph, describe our algorithm for selecting the set $L$ given a threshold $\tau$. First, we initialize $L = \emptyset$. Then, while the $s$-$t$ min-cut in $\widehat{G}_{L,\tau}$ is not $n \cdot \tau$ and $L \leq \lceil 2 \cdot \log_2 nW + 2 \rceil \cdot k$, we iterate over all vertices $v \in V \setminus L$ and compute $mincut_{\widehat{G}_{L \cup \{v\}}}(s,t)$. We then choose the vertex that achieves the largest min-cut and add it to $L$. See also Algorithm 1.

**Analysis.** In this section, we show that Algorithm 1 solves the T-GLS problem (Definition 3.3).

**Lemma 5.1.** *For a graph $G = (V, E, \boldsymbol{w})$ such that $\max_{L \subset V : |L| \leq k} \Psi(L) \geq \tau$, the greedy algorithm finds a set of size $L'$ at most $O(\log n) \cdot k$ such that $\Psi(L') \geq \tau$.*

---

**Algorithm 1** T-GLS($G, \tau, k$)

**Input:** Graph $G = (V, E, \boldsymbol{w})$ with $n = |V|$ and $\sum_{e \in E} \boldsymbol{w}(e) = W$, threshold $\tau \in \mathbb{R}$, and budget $k \in \{1, \ldots, n\}$.
$L \leftarrow \emptyset$; $s \leftarrow \lceil 2 \cdot \log_2 nW + 2 \rceil$
**while** $|L| \leq s \cdot k$ and $mincut_{\widehat{G}_{L,\tau}}(s,t) < n \cdot \tau$ **do**
  $u \leftarrow \arg\max_{v \in V \setminus L} mincut_{\widehat{G}_{L \cup \{v\}}}(s,t)$
  $L \leftarrow L \cup \{u\}$
**end while**
**return** $L$

---

*Proof.* We first observe that the value of the maximum flow in the graph only increases throughout the execution of the greedy algorithm by Observation 4.2. Therefore, the gap $g$ between the current flow and $n \cdot \tau$ only decreases.

Throughout, there is a flow supported on at most $k$ extra edges with gap 0 by $\max_{L \subset V : |L| \leq k} \Psi(L) \geq \tau$ and Claim 4.5. Any such flow routes at least $g$ flow over these $k$ edges. Fix an integral such flow $\boldsymbol{f}$. By the pigeonhole principle, it routes at least $\lceil g/k \rceil$ over one of the extra edges. Fixing such an edge, and removing all flow paths from $s$ to $t$ crossing other newly added edges shows that there exists an individual edge that decreases the gap by at least $\lceil g/k \rceil$.

Now, consider $k$ consecutive steps of the algorithm with initial gap $g$. As long as the gap has not halved, it gets decreased by at least $g/2k$ every step by the argument presented in the previous paragraph with gap $g/2$. Therefore, the gap gets halved after $k$ steps.

Since the gap is an integer, we conclude that running the algorithm for $\lceil 2 \log(nW) + 2 \rceil \cdot k$ steps achieves gap 0 because the gap gets halved every $k$ steps and is at most $nW$ at the start.

Together with Corollary 4.8, this concludes the proof. □

**Theorem 5.2.** *There is an algorithm that solves the T-GLS problem (Definition 3.3) for a graph with $n$ vertices and $m$ edges of total weight $W$ with slack $s = O(\log n\tau)$ that runs in time $O(nk \log nW) \cdot M(2n + m, 2W + 1)$ where $M(2n + m, 2W + 1)$ denotes the time it takes to compute a maximum flow on a graph with $2n+m$ edges with capacities bounded by $2W + 1$.*

*Proof.* By Lemma 5.1, we are guaranteed to find a set of size at most $O(\log n\tau) \cdot k$ whenever a set of size $k$ exists. The runtime bound directly follows from the description of our algorithm. □

**Extremely large edge weights.** The guarantee on $s$ in Theorem 5.2 becomes meaningless when $\tau$ grows very large, which can happen on graphs with extremely large edge weights $\boldsymbol{w}(e) \gg n^{O(1)}$. Fortunately, we can avoid this issue by slightly relaxing the guarantee on $\Psi(L')$ and rounding edge weights.

**Definition 5.3** (Relaxed Thresholded Graph Label Selection (RT-GLS)). Given a graph $G = (V, E, \boldsymbol{w})$, a budget $k \leq |V|$ and a threshold $\tau \in \mathbb{R}_{>0}$, the thresholded graph label selection problem with slack $s$ and error $\epsilon$ is to either

- certify that $\max_{L \subseteq V : |L| \leq k} \Psi(L) < \tau$ or

- output a set $L$ with $|L| \leq s \cdot k$ such that $\Psi(L) \geq (1 - \epsilon) \cdot \tau$.

Given error $\epsilon$, the magnitude of the relevant edge weights can be reduced drastically.

**Theorem 5.4.** *There is an algorithm for RT-GLS (Definition 5.3) for a graph with $n$ vertices and $m$ edges with slack $s = O(\log n/\epsilon)$ that runs in time $O(nk \log \frac{n}{\epsilon}) \cdot M(2n + m, W')$ where $M(2n + m, W')$ denotes the time it takes to compute a maximum flow on a graph with $2n + m$ edges with weights bounded by $W' = O(n^2/\epsilon^2)$.*

*Proof.* We first remove all edges with weight less than $\epsilon\tau/(2n)$ from the graph. For any cut, these edges contribute at most $\epsilon\tau/2$ mass. Edges with weight larger than $2\tau \cdot n$ can be reduced to $2\tau \cdot n$ without affecting the solution, since any cut that includes such edges has ratio at least $2\tau$. This reduces all the edge weights to the range $[\epsilon\tau/(2n), 2\tau \cdot n]$ Dividing all the edges by $\epsilon\tau/10$ and rounding up again introduces at most error $\epsilon\tau/2$ when scaling back to the original magnitude. We then obtain our result via running the algorithm from Theorem 5.2 on the graph with the rounded down edge weights and scaling up the solution. □

We conclude this section with our main result.

**Theorem 5.5.** *There is an algorithm that solves the GLS problem (Definition 3.2) for a graph with maximum integral edge weight $U$ with*

1. *slack $O(\log nU)$ and error $0$ in $O(nk)(n+m)^{1+o(1)} \cdot \log^3 U$ time and;*

2. *slack $O(\log n/\epsilon)$ and error $\epsilon > 0$ in $O(nk) \cdot (n + m)^{1+o(1)} \cdot \log^3 \epsilon^{-1}$ time.*

*Proof.* The first item directly follows from Theorem 5.2 and Lemma 3.4 with current algorithms for computing maximum flows (Chen et al., 2023; van den Brand et al., 2024) where we notice that $\Psi(L) \leq U$ for $L \neq V$. The second item follows by Theorem 5.4 and a slightly refined threshold search, where we first find the largest edge weight $\boldsymbol{w}(e)$ such that the threshold $\boldsymbol{w}(e)/n$ can be achieved, but $2n \cdot \boldsymbol{w}(e)$ cannot by binary search. Such an edge is guaranteed to exist since the optimal threshold has to fall into that range for some edge with largest weight in a witness cut. We may assume $\epsilon < 1$ with no loss of generality. This implies that once we found such a range we can perform binary search with small enough $\epsilon' < \epsilon/10$ until we find a close enough approximation. The runtime follows directly. □

This directly implies Theorem 1.1 as a corollary.

*Proof of Theorem 1.1.* Directly follows from the first item of Theorem 5.5. □

# 6. Hardness of Approximation

In this section, we show hardness of approximation. We first relate our problem to independent sets.

**Lemma 6.1.** *Assume to be given a $\Delta$-regular unweighted graph $G = (V, E)$, and threshold $\tau = \Delta$. Then $\Psi(L) \geq \Delta$ if $V \setminus L$ is an independent set, and $\Psi(L) \leq (2\Delta - 2)/2$ otherwise.*

*Proof.* If two vertices $u, v$ in $V \setminus L$ are be adjacent, they form a cut $C = \{u, v\}$ of value $\Psi_C = (2\Delta - 1)/2 < \Delta$ that does not include a vertex in $L$. Otherwise, $V \setminus L$ is an independent set and we obtain $\Psi(L) \geq 3$ directly. □

See Figure 3 for an illustration of our reduction from maximum independent set on regular graphs.

*Proof of Theorem 1.2.* The maximum independent set problem is NP-hard even on graphs which are 3-regular (Fleischner et al., 2010; Alimonti & Kann, 2000). Therefore, the theorem follows from Lemma 6.1, since we could otherwise test for the existence of independent sets of size $n - k$ for arbitrary $k$. □

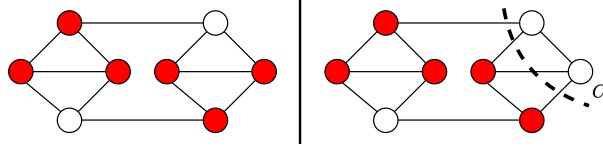

*Figure 3.* This figure shows two graphs with vertices in $L$ depicted in red. On the left hand side, the non-labeled vertices form an independent set. Therefore, $\tau = \Delta = 3$. On the right hand side, they do not. Choosing two connected vertices proves that there is a cut of sparsity $4/2 = 2$.

## 7. Experiments

To complement our theoretical results, we run a proof-of-concept experiment where we compare our algorithm practical performance with the previous algorithms of (Guillory & Bilmes, 2009) and (Cesa-Bianchi et al., 2010). We recall that the algorithm of (Guillory & Bilmes, 2009) is heuristic, whereas the algorithm of (Cesa-Bianchi et al., 2010) guarantees constant slack on unweighted trees. To handle arbitrary unweighted graphs, the authors suggest to sample a random spanning tree of the graph before running their algorithm.

**Experimental setup.**    For all experiments, we run the algorithm of (Guillory & Bilmes, 2009) for maximizing $\Psi(\cdot)$ and the algorithm of (Cesa-Bianchi et al., 2010) 10 times and report the mean and standard deviation for the achieved ratio $\Psi(L)$. Since our algorithm is deterministic, we simply report the ratio. We don't allow any slack in the budget $k$ when evaluating the algorithms.[5]

We first evaluate the algorithms on two simple graphs that cause the previous heuristic algorithms to fail. The first such graph is the star on $50$ vertices, which we denote as S(50). The second graph 3C(10, 30, 10) is obtained by taking a path consisting of 3 vertices, and replacing the central vertex with a clique of size 30 and the two degree one vertices with a clique of size 10 each.

We then explore the full trade-off between budget $k$ and $\Psi(L)$ for the Davis southern woman graph (Davis et al., 1941). This is a small graph modeling a social circle consisting of 32 vertices.

Finally, we run experiments on two real world graphs from the Stanford Network Analysis Project (SNAP) (Leskovec & Krevl, 2014). The graph ca-GrQc models the Arxiv GR-QC (General Relativity and Quantum Cosmology) collaboration network and ego-Facebook is a Facebook friend network. Since ca-GrQc is not connected, we run the algorithms on

the largest connected component ($4158$ vertices).

**Results.**    We report the results for the simple graphs in Table 1. For S(50) our algorithm and (Cesa-Bianchi et al., 2010) always choose the center of the star, but (Guillory & Bilmes, 2009) often chooses 3 leaves. For 3C(10, 30, 10), our algorithm and the algorithm of (Guillory & Bilmes, 2009) always chooses one vertex from each clique, which the algorithm of (Cesa-Bianchi et al., 2010) fails to do.

We plot the trade-off between budget and ratio for the Davis southern woman graph in Figure 4.

Finally, we report he results for the SNAP graphs in Table 2.

In all the experiments we notice that our algorithm obtains better performance than previous work.

We remark that when the size of the label set approaches $n$, our algorithm gets outperformed by (Guillory & Bilmes, 2009). A simple reason for this behavior is that for a star graph our algorithm always includes the center, whereas the algorithm of (Guillory & Bilmes, 2009) sometimes picks all the leaves. For $k = n - 1$, it is advantageous to pick all the leaves (See Figure 1).[6]

We point the interested reader to Appendix C for additional experiments on synthetic graphs.

| Interpretable Example Graphs for $k = 3$ | | |
|---|---|---|
| Graph | S(50) | 3C(10, 30, 10) |
| (Guillory & Bilmes, 2009) | 0.16±0.28 | 1.0 ± 0.0 |
| (Cesa-Bianchi et al., 2010) | 1.0 ± 0.0 | 0.1 ± 0.03 |
| **Ours** | **1.0** | **1.0** |

*Table 1.* Experimental results for interpretable example graphs rounded to two digitis after the decimal point.

## 8. Generalizing to Vertex Importances

If the data set contains outliers, our methods are highly encouraged to label them. (Guillory & Bilmes, 2009) point out that this leads to sub-par performance on many real-world data sets. We present a natural remedy for this issues by generalizing the GLS problem to vertex importances $f : V \mapsto \mathbb{R}_{\geq 0}$. The vertex importance function allows removing weight from outliers. A particularly natural function is $f(\cdot) = \deg(\cdot)$. Then, sparsity corresponds to conductance and vertices that are very dissimilar to all other vertices receive a low importance.

**Definition 8.1** (Importance Sparsity)**.** For a graph $G =$

---

[5]We provide the code used for running the experiments under https://github.com/mesimon/graph_label_selection.

[6]With a small budget, labeling vertices with high degree is generally advantageous since they cover a large part of the graph. When the budget approaches $n$, these are vertices for which one can learn a lot without labeling them, and it therefore becomes advantageous to label their neighborhood instead.

| Graph | ca-GrQc ($n = 4158$) | | | ego-Facebook ($n = 4039$) | | |
|-------|--------|--------|--------|--------|--------|--------|
| Budget | $k = 10$ | $k = 50$ | $k = 100$ | $k = 10$ | $k = 50$ | $k = 100$ |
| GB | 0.015±0.006 | 0.066±0.011 | 0.132±0.011 | 0.076±0.008 | 0.396±0.055 | 0.777±0.059 |
| BGVZ | 0.054±0.006 | 0.062±0.000 | 0.062±0.000 | 0.037±0.010 | 0.300±0.376 | 0.831±0.339 |
| **Ours** | **0.091** | **0.215** | **0.333** | **1.0** | **1.0** | **1.080** |

*Table 2.* This table compares the ratio $\Psi(\cdot)$ achieved by GB (Guillory & Bilmes, 2009), BGVZ (Cesa-Bianchi et al., 2010) and our algorithm for several budgets on two real world graphs from the Stanford Network Analysis Project (Leskovec & Krevl, 2014). We round to three digits after the decimal point.

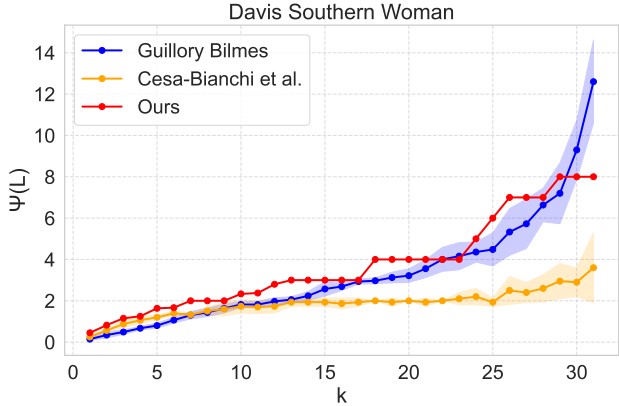

*Figure 4.* The full trade-off between the budget $k$ and the achieved threshold for the Davis southern woman graph (Davis et al., 1941).

$(V, E, \boldsymbol{w})$, a positive vertex function $f : V \mapsto \mathbb{R}_{\geq 0}$ and a cut $\emptyset \subset C \subset V$, we let

$$\Psi_C^f \overset{\text{def}}{=} \frac{\boldsymbol{w}(C, V \setminus C)}{f(V)}$$

where $f(V) \overset{\text{def}}{=} \sum_{v \in V} f(v)$. Furthermore, we let $\Psi_\emptyset^f \overset{\text{def}}{=} \infty$, and $\Psi_V^f \overset{\text{def}}{=} 0$.

Notice that for the constant function $f(\cdot) = 1$, we obtain $\Psi_C^f = \Psi_C$ for all $C$ and Definition 8.1 therefore generalizes Definition 3.1. We then define the I-GLS problem, which is a generalization of the GLS problem Definition 3.2.

**Definition 8.2** (I-GLS). Given a graph $G = (V, E, \boldsymbol{w})$, a positive vertex function $f : V \mapsto \mathbb{R}_{\geq 0}$ and a budget $k \leq |V|$, we let

$$\Psi^f(L) \overset{\text{def}}{=} \min_{C \subseteq V \setminus L} \Psi_C^f.$$

The I-GLS objective is then given by

$$\max_{L \subseteq V : |L| \leq k} \Psi^f(L).$$

We call any solution $L'$ such that $|L'| \leq s \cdot k$ and

$$\Psi^f(L') \geq \max_{L \subseteq V : |L| \leq k} \Psi^f(L) \qquad (1)$$

a solution with slack $s \in \mathbb{R}_{\geq 0}$.

Next, we show that our algorithm can be extended to also solve the I-GLS problem directly.

**Theorem 8.3.** *Assume to be given a graph $G = (V, E, \boldsymbol{w})$ alongside a vertex function $f : V \mapsto \mathbb{R}_{\geq 0}$ and a budget $k$, such that $\boldsymbol{w}(e) \in \{1, 2, \ldots |V|^{O(1)}\}$ and $f(v) \in \{1, 2, \ldots |V|^{O(1)}\}$ for all $e \in E$ and $v \in V$ respectively. Then, there exists an algorithm that returns a set $L'$ such that*

- *$|L'| \leq O(\log |V|)$ and*

- *$\Psi(L') \geq \max_{L \subset V : |L| \leq k} \Psi(L)$.*

*The algorithm runs in time $k \cdot |V| \cdot (|V| + |E|)^{1+o(1)}$ where $o(1)$ goes to 0 as $|V|$ approaches infinity.*

*Remark* 8.4. We assume that the similarity vector $\boldsymbol{w}$ and the vertex importances $f(\cdot)$ are polynomially bounded in $|V|$ for ease of presentation. Slightly relaxing the objective allows removing this assumption analogously to Theorem 5.5.

Since the proof of Theorem 8.3 directly follows the outline of the proof of Theorem 1.1 we only sketch the alterations and include a full proof in Appendix B for completeness.

**Proof sketch of Theorem 8.3.** To incorporate the importance function $f : V \mapsto \mathbb{R}_{\geq 0}$ into the flow gadget from Definition 4.1, we simply multiply the edge weight of $(s, v)$ with the importance $f(v)$ of vertex $v$ for every edge adjacent to $s$. We furthermore increase the weight of the edges $(v, t)$ for $v \in L$ sufficiently such that they are still effectively infinite.

Then, by following the proof outline of Claim 4.5 and Claim 4.6, we directly observe that this altered gadget decides the threshold version of Definition 8.2. This again enables us to solve the problem using binary search.

## 9. Conclusion

We present the first approximation algorithm for maximizing the graph label selection objective $\Psi(L)$ in a general graph with theoretical guarantees, and extend it to the case where

not all data points are equally important. We also show that maximizing $\Psi(L)$ to high accuracy is NP hard. Finding better resource augmented algorithms as well as finding approximation algorithms for $\Psi(L)$ without relaxing the cardinality constraint remain interesting open problems.[7] Furthermore, it is interesting whether our lower bound can be strengthened or extended to more natural graph instances.

Although our algorithm is relatively practical, it runs in time much larger than linear. This prevents us from scaling to large real world graphs. However, it is encouraging that much faster approximation algorithms for the densest subgraph problem have been developed (Chekuri et al., 2022). It is an interesting research direction to understand whether similar techniques could drastically improve the scalability for graph label selection, which would enable gathering experimental results on a larger scale. We remark that submodular optimization is highly sensitive to adversarial noise (Hassidim & Singer, 2017), and therefore approximate flow oracles as developed in (van den Brand et al., 2024) cannot be used directly to speed up our algorithm.[8]

## Acknowledgments

Simon Meierhans is supported by a Google PhD Fellowship and grant no. 200021 204787 of the Swiss National Science Foundation.

## Impact Statement

This paper presents work whose goal is to advance the field of Machine Learning. The nature of our work is mainly theoretical, and thus our insights might find use in various areas.

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

## A. Submodularity

Submodular functions are an important function class in combinatorial optimization. We first state one of multiple equivalent definitions of submodularity.

**Definition A.1.** A function $f : 2^A \mapsto \mathbb{R}$ is called submodular if for all $X, Y \in A$ and $a \in A$ such that $X \subseteq Y$ we have $f(X - \{a\}) - f(X) \geq f(Y - \{a\}) - f(Y)$.

Intuitively, this corresponds to the notion of diminishing returns. We re-consider the simple example from Figure 1 in Figure 5, and observe that $\Psi(X \cup \{a\}) - \Psi(X) = 0$ whereas $\Psi(Y \cup \{a\}) - \Psi(Y) = 2$.

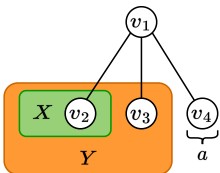

*Figure 5*. A star graph with 3 leaves and sets $X = \{v_2\}$, $Y = \{v_2, v_3\}$, and $a = v_4$ is a counterexample to $\Psi(\cdot)$ being submodular. We have $\Psi(X) = \Psi(Y) = 1$, as well as $\Psi(X \cup \{a\}) = 1$. But $\Psi(Y \cup \{a\}) = 3$, which implies $\Psi(X \cup \{a\}) - \Psi(X) < \Psi(Y \cup \{a\}) - \Psi(Y)$. Therefore the function $\Psi(\cdot)$ is not submodular.

Next, we define supermodularity.

**Definition A.2.** A function $f : 2^A \mapsto \mathbb{R}$ is called supermodular, if $-f$ is submodular.

Unfortunately, the function $\Psi(\cdot)$ is also not supermodular. To show this, we consider a line graph on 4 vertices in Figure 6.

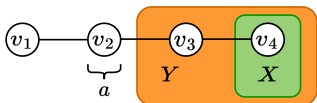

*Figure 6*. We let $Y = \{v_3, v_4\}$ and $X = \{v_4\}$. Then, we have $\Psi(X) = 1/3$ and $\Psi(Y) = 1/2$. For $a = v_2$, we obtain $\Psi(X \cup \{a\}) = 1$, and $\Psi(Y \cup \{a\}) = 1$. We obtain $\Psi(X \cup \{a\}) - \Psi(X) > \Psi(Y \cup \{a\}) - \Psi(Y)$, which shows that the function $\Psi(\cdot)$ is not supermodular either.

In contrast to the function $\Psi(\cdot)$, the function $f : 2^V \mapsto \mathbb{R}$ defined by $f(A) = \mathrm{mincut}_{\widehat{G}_{A,\tau}}(s, t)$ (See Definition 4.1) is submodular. Recall that every vertex $v \in A$ corresponds to adding an edge $(v, t)$ to $\widehat{G}_{A,\tau}$. We can observe that all the extra flow that can be routed when adding the edge $(a, t)$ to $\widehat{G}_Y$ can also be routed when adding $(a, t)$ to $\widehat{G}_X$.

## B. Proof of Theorem 8.3

To prove Theorem 8.3, we first introduce a generalization of our flow gadget.

**Definition B.1** ($f$-Flow Graph). Assume to be given an undirected graph $G = (V, E, \boldsymbol{w})$, a function $f : V \mapsto \mathbb{R}_{\geq 0}$, a set $L$ and a threshold $\tau$. Let $W$ denote the sum of the edge weights of $G$. We then construct a graph $\widehat{G}^f_{L,\tau}$ as follows.

- (Vertex set). The vertex set of $\widehat{G}^f_{L,\tau}$ is $V \cup \{s, t\}$.

- (Copy). Every edge $e \in E$ is present in $\widehat{G}^f_{L,\tau}$ with the same weight.

- (Source). For every vertex $v \in V$, the graph $\widehat{G}^f_{L,\tau}$ contains an edge $(s, v)$ with weight $f(v) \cdot \tau$.

- (Sink). For every vertex $l \in L$, the graph $\widehat{G}^f_{L,\tau}$ contains an edge $(l, t)$ with weight $W + f(v) \cdot \tau + 1$.

We then show analogous claims to Claim 4.5 and Claim 4.6.

**Claim B.2.** *If* $mincut_{\widehat{G}^f_{L,\tau}}(s,t) < f(V) \cdot \tau$, *then* $\Psi^f(L) < \tau$.

*Proof.* Fix some $s$-$t$ min-cut $C$ in $\widehat{G}^f_{L,\tau}$. As in the proof of Claim 4.5, we have that $C \cap V \neq \emptyset$ as otherwise $mincut_{\widehat{G}^f_{L,\tau}}(s,t) = f(V) \cdot \tau$. We then have

$$\boldsymbol{w}(C, V \setminus C) + f(V \setminus C) \cdot \tau = mincut_{\widehat{G}_{L,\tau}}(s,t)$$
$$< \tau \cdot f(V)$$

and obtain $\frac{w(C,V\setminus C)}{f(C)} < \tau$ by reordering and $f(V) - f(V \setminus C) = f(C)$. This concludes the proof of the claim. $\square$

**Claim B.3.** *If* $\Psi(L) < \tau$, *then* $mincut_{\widehat{G}_{L,\tau}}(s,t) < f(C) \cdot \tau$.

*Proof.* Consider a set $C \subseteq V$ that certifies $\Psi(L) < \tau$, i.e. a set $C$ such that $\boldsymbol{w}(C, V \setminus C) < \tau \cdot f(C)$. Then the cut $\{s\} \cup C$ has value $f(V)\tau + \boldsymbol{w}(C, V \setminus C) - \tau \cdot f(C) < f(C) \cdot \tau$. This concludes the proof of the claim. $\square$

*Proof of Theorem 8.3.* We first show that there is an algorithm that either

- certify that $\max_{L \subseteq V : |L| \leq k} \Psi^f(L) < \tau$ or

- output a set $L$ with $|L| \leq s \cdot k$ such that $\Psi^f(L) \geq \tau$.

using $|L|$ calls to a maximum flow oracle. This generalization of the T-GLS problem (Definition 3.3) can be solved by adapting our flow gadget to $\widehat{G}^f_{L,\tau}$ (Definition B.1). By Claim B.2 and Claim B.3 the set $L$ achieves threshold $\tau$ if and only if the $s$-$t$ min-cut in $\widehat{G}^f_{L,\tau}$ is $\tau \cdot f(V)$.

By an analogous argument to the proof of Lemma 5.1, the greedy algorithm with slack $O(\log n)$ finds a solution obtaining threshold $\tau$ if a size $k$ solution achieving threshold tau exists.

The theorem follows from standard binary search analogous to the proof of Lemma 3.4. $\square$

## C. Additional Experiments on Synthetic Graphs

We explore the trade-off between budget and objective for various Watts-Strogatz random graphs with 50 vertices. We let Watts-Strogatz$(50, d, p)$ refer to the Watts-Strogatz graph with average degree $d$ and re-linking probability $p$ (Watts & Strogatz, 1998) and perform experiments for the graphs Watts-Strogatz$(50, 4, 0.1)$, Watts-Strogatz$(50, 4, 0.2)$, Watts-Strogatz$(50, 8, 0.1)$ and Watts-Strogatz$(50, 8, 0.2)$.

For each of these graphs, we run our algorithm as well as the algorithms of (Guillory & Bilmes, 2009) and (Cesa-Bianchi et al., 2010) for all budgets $k$ and plot the mean and standard deviation after 10 experiments for the algorithms involving randomization.

The results of our experiments are shown in Figures 7 to 10. We again observe that our algorithm outperforms the previous algorithms.

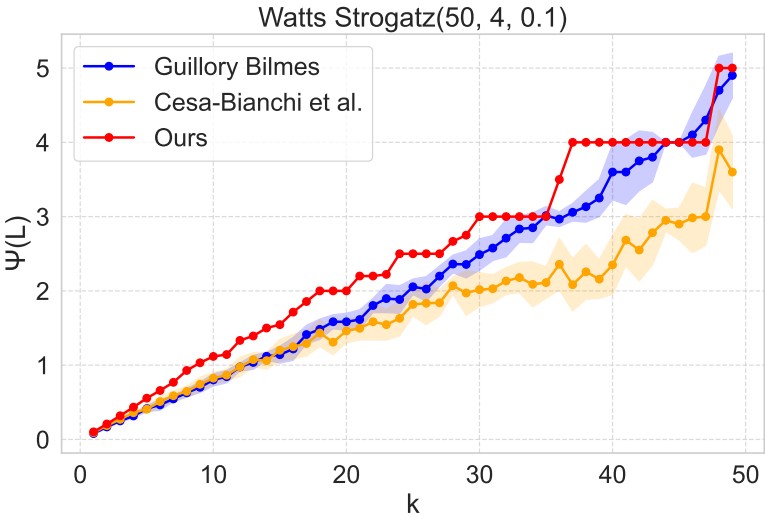

*Figure 7.* Trade-off between the budget $k$ and the achieved threshold for a Watts-Strogatz random graph with degree $4$ and rewiring probability $p = 0.1$ on $50$ vertices (Watts & Strogatz, 1998).

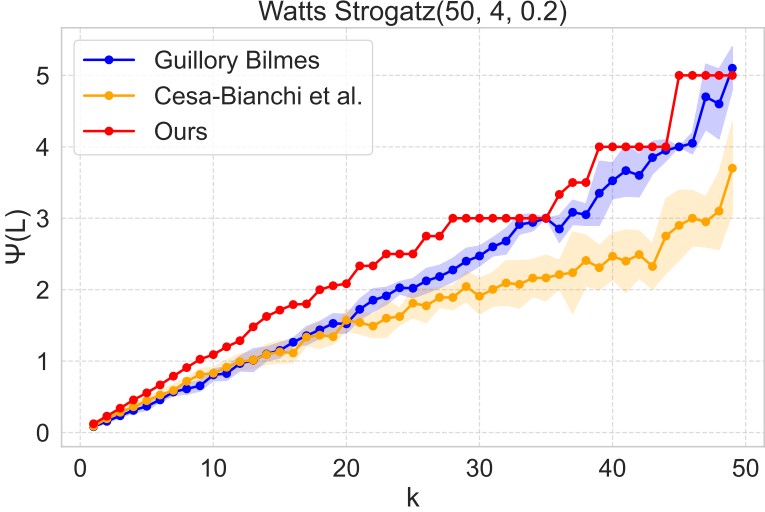

*Figure 8.* Trade off between the budget $k$ and the achieved threshold for a Watts-Strogatz random graph with degree $4$ and rewiring probability $p = 0.2$ on $50$ vertices (Watts & Strogatz, 1998).

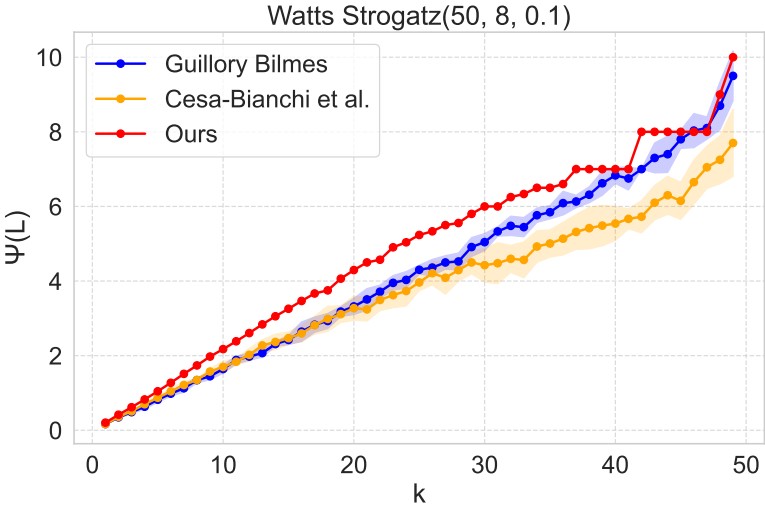

*Figure 9.* Trade-off between the budget $k$ and the achieved threshold for a Watts-Strogatz random graph with degree $8$ and rewiring probability $p = 0.1$ on $50$ vertices (Watts & Strogatz, 1998).

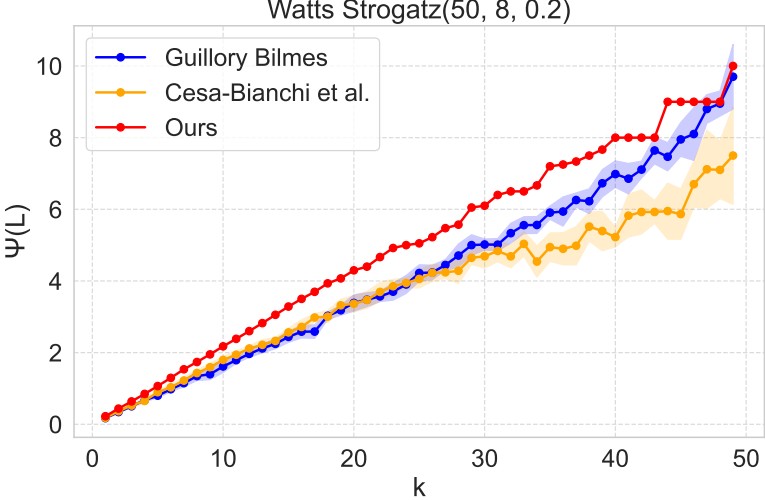

*Figure 10.* Trade-off between the budget $k$ and the achieved threshold for a Watts-Strogatz random graph with degree $8$ and rewiring probability $p = 0.2$ on $50$ vertices (Watts & Strogatz, 1998).

