# OpenReview forum: "Algorithms and Hardness for Active Learning on Graphs"
_ICML.cc/2025/Conference — ICML 2025 poster_

### Official Review · Reviewer_gCbm · 2025-03-09

**Overall Recommendation:** 4

**Summary:**

The authors study a "Graph Label Selection" (GLS) problem of Blum and Chawla and Guillory and Bilmes, which, given a graph G=(V,E) and parameter k, asks the learner to find a subset |L|=k of vertices maximizing the unnormalized min-cut outside of $L$:

$\Psi(L)=\min_{C \subset V \setminus L} \frac{e(C,V\setminus C)}{|C|}$

This problem is related to active learning on graphs, where finding a subset with large $\Psi(L)$ leads to a generalization error upper bound for labeling the vertices of the graph assuming labels between points with edges are similar on average.

The authors prove two main results:

1) Via reduction to independent set, it is NP-Hard to determine whether $\Psi(L) \leq 2$ or $\Psi(L) \geq 3$

2) There is a poly time algorithm for GLS on any graph with *log(|V|) slack*, i.e. it is possible to find L of size $O(k\log(|V|))$ such that $\Psi(L) \geq \max_{|L'|=k} \Psi(L')$

The authors also give some experimental evidence that executing their algorithm with no slack is as good or better than prior heuristics on many datasets.

The main new technical contribution of the work is an algorithm for building such $L$ by equating the computation of $\Psi(L)$ to mincut/maxflow in a closely related auxiliary graph. The authors then show that by greedily finding the vertex which maximizes the auxiliary min-cut, one iteratively halves the gap to the true $\Psi(L)$ value every k steps, resulting in the $k\log(|V|)$ slack bound.

**Claims And Evidence:**

Yes

**Essential References Not Discussed:**

Not to my knowledge.

**Experimental Designs Or Analyses:**

No.

**Methods And Evaluation Criteria:**

Yes

**Other Comments Or Suggestions:**

Typo in Lemma 6.1 proof, should say Psi(L) \geq Delta

“he results”

**Other Strengths And Weaknesses:**

It would be helpful if the authors included more discussion how the studied problem is actually related to learning on graphs. From my understanding looking up previous works, finding $L$ with large $\Psi(L)$ corresponds to a learning strategy of Blum and Chawla getting good generalization error assuming that most pairs of vertices with edges are given the same label. This motivates the GLS (with slack) problem, since it essentially shows that if one can achieve low error using $k\log(|V|)$ label queries assuming the $k$-GLS value is small.

The algorithm and proofs presented in the paper are simple and natural, which I view as a strength, especially since previously algorithms were only known for trees. Giving an algorithm for general graphs therefore seems like a big advance on the problem, even if prior methods for trees had constant vs logarithmic slack.

I find the hardness result presented a bit weak. While it is formally true the result implies 3/2-approximating GLS is NP-hard, it is only for a very specific setting of parameters. Could one hope to achieve an algorithm with additive error? What about when the GLS value is large (indeed generally it seems one might hope for this in the downstream learning application?). Can one prove hardness for learning GLS with constant slack for general graphs?

**Questions For Authors:**

N/A

**Relation To Broader Scientific Literature:**

The key contribution of this paper is a new algorithm for the graph label selection problem (itself related to active learning on graphs) for *arbitrary* underlying graphs with $O(\log(|V|))$ slack. Previously algorithms (with slack) were only known for trees, with heuristic methods suggested for general graphs. The authors also prove the first NP-hardness result for GLS with no slack, though this aspect is not particularly surprising given typical hardness of such graph-based combinatorial problems.

**Theoretical Claims:**

Yes. The proofs seem fine.

---

> ### Author Rebuttal · Authors · 2025-03-30
>
> Dear reviewer gCbm,
>
> We thank you for your work reviewing our paper, we will address your points in the final version. Here we will focus on discussing  the points you raised.
>
> 1) Motivation of GLS: Thanks for the suggestion. We will include a discussion of the motivation behind the GLS objective in our article to better motivate the question.
>
> 2) Hardness of approximation: Hardness of approximation results that are either stronger or capture more parameter regimes are certainly interesting questions for future research. Given the problem definition, it is plausible that the problem might inherit the hardness of sparsest cut, although we were not able to prove it. An additive approximation may also be possible.
>
> 3) Thank you for pointing out some typos in our article.

---

### Official Review · Reviewer_3ZDj · 2025-03-14

**Overall Recommendation:** 4

**Summary:**

The authors propose an approximation algorithm (or, more specifically, a resource augmented algorithm) for the graph label selection problem (GLS). GLS is an abstraction of the active learning task of selecting a small set of data points to label out of a pool of unlabeled data.

The main contribution of the authors is an algorithm with theoretical guarantees on the quality of the solution for *general graphs*, whereas previous works propose approximation algorithms on restricted types of graphs. Moreover, the authors present a proof on the hardness of approximation for GLS in general graphs.

## update after rebuttal
Not much to add. After reading the other reviews, I maintain my original score and review that this paper is above the acceptance bar.

**Claims And Evidence:**

Yes.

**Essential References Not Discussed:**

No.

**Experimental Designs Or Analyses:**

Yes. I read through the description of the datasets and the metrics. No issues identified.

**Methods And Evaluation Criteria:**

Yes.

**Other Comments Or Suggestions:**

Proofread Section 6 and Section 7. The writing is sloppy compared to the rest of the paper.

**Other Strengths And Weaknesses:**

None

**Questions For Authors:**

In Figure 4, the heuristic of Guilroy and Bilmes has better performance than your proposed method for k >= 30. This is not acknowledged or mentioned in the text. Do you have any explanation as to why the heuristic is better in this case?

**Relation To Broader Scientific Literature:**

The key contributions of the paper are clearly novel and provide new insights on the hardness of GLS. Since GLS is a simplified version of the active learning task, the paper contributes theoretical results on the hardness of active learning. This is one core task in machine learning / data science.

**Theoretical Claims:**

Yes. I checked all the theorems and lemmas to the best of my ability. The theoretical results are the bulk of the contribution.

There seem to be typos / inconsistencies in Section 6 and Lemma 6.1. This whole section seems somewhat carelessly written compared to the rest of the paper. In particular, halfway through the proof of Lemma 6.1, the authors switch from talking about $\Delta$-regular graphs to the specific case of $\Delta=3$.

---

> ### Author Rebuttal · Authors · 2025-03-30
>
> Dear reviewer 3ZDj,
>
> We thank you for your work reviewing our paper. Here we focus on responding to your question and the points you raised.
>
> 1) ``In Figure 4, the heuristic of Guilroy and Bilmes has better performance than your proposed method for $k \geq 30$. This is not acknowledged or mentioned in the text. Do you have any explanation as to why the heuristic is better in this case?": This behaviour appears because when k approaches the number of vertices, it is no longer necessarily beneficial to select vertices which have a lot of neighbors.
> This  can be illustrated well by considering a star graph on say $n$ vertices with budget $k = n - 1$. Counterintuitively, The best solution for this example is to pick all the leaves. This solution achieves score $n - 1$.
> Since our algorithm follows a greedy approach, it will always pick the star center and then all but one of the leaf vertices. This only achieves score $1$. The heuristic of Guilroy and Bilmes on the other hand, as long as the star center is unlabeled,  will always sample a vertex from a set that contains the star center, but also all but one of the currently unlabeled leaves. Some simple calculations show that this means that the algorithm actually has a reasonably large chance of around $1/\log n$ to not include the star center and obtain the optimal solution. Since the gap between the two solutions is $n - 1$, this very significant influences the expectation in this degenerate scenario. We will add a discussion of this phenomenon to our article.
>
> 2) We will proofread and edit section 6 and 7 to improve the writing in our article.

---

> > ### Comment · Reviewer_3ZDj · 2025-04-03
> >
> > I thank the authors for addressing my comments. I will maintain my original score.

---

### Official Review · Reviewer_qHp3 · 2025-03-14

**Overall Recommendation:** 3

**Summary:**

This paper tackles the problem of *active learning on graphs* under a label smoothness assumption. The authors study how to select a set of $k$ labeled vertices in a graph such that the labels can best predict all other vertices’ labels. The core contributions are two-fold: **(1)** a new **approximation algorithm** (with resource augmentation) that is the first to offer theoretical guarantees on general graphs, and **(2)** a **hardness result** proving that no efficient algorithm can achieve high accuracy in general without relaxing the problem. Specifically, they present an algorithm that, using an $O(\log n)$ larger label budget, achieves an objective value at least as good as the optimal set of $k$ labels. In complement, they prove it is NP-hard to distinguish whether the optimal value is small (2) or moderately larger (3), implying a constant-factor hardness of approximation for the exact budget-$k$ problem. Beyond theory, the paper includes proof-of-concept experiments on both synthetic graphs and real-world graph datasets, which indicate that the proposed method outperforms prior heuristics on this task.

**Claims And Evidence:**

The paper’s claims are generally well-supported by theoretical arguments and empirical validation. The **approximation guarantee** (the first efficient algorithm with theoretical bounds for general graphs) is clearly stated and backed by rigorous proofs. The **hardness result**, which demonstrates NP-hardness of achieving high accuracy for the considered objective, is convincingly justified by a known reduction from a well-studied NP-hard problem. Experimentally, the authors substantiate their claim of improved performance by comparing the proposed algorithm to baselines (Guillory & Bilmes, 2009; Cesa-Bianchi et al., 2010) on both synthetic and real-world graphs, consistently showing superior results. The authors responsibly qualify their contributions by explicitly noting limitations such as resource augmentation and computational complexity, making their claims balanced and credible.

Guillory & Bilmes, 2009: Label selection on graphs.
Cesa-Bianchi et al., 2010: Active learning on trees and graphs.

**Essential References Not Discussed:**

No essential references appear missing. Literature coverage seems thorough, and relevant works are adequately cited.

**Experimental Designs Or Analyses:**

The proposed method uses a flow-based formulation inspired by the **densest subgraph** problem, constructing a flow gadget to guide greedy vertex selection. Although the original label selection objective ($\Psi(L)$) is neither submodular nor supermodular, the authors identify a submodular surrogate measure (flow increase), enabling standard greedy approximation techniques. They combine this with binary search for threshold estimation, clearly outlining assumptions and deferring details to appendices.

The evaluation methodology (theoretical approximation ratio, empirical objective values) is suitable for the paper's goals. Empirical results on synthetic corner-case graphs and two real-world graphs from SNAP consistently demonstrate better performance over baseline methods (Guillory & Bilmes, 2009; Cesa-Bianchi et al., 2010). A notable limitation is the relatively small scale of experiments (max ~4k nodes), acknowledged by authors due to computational complexity. Nonetheless, the experiments convincingly support the claims within the tested scope. Reproducibility is reasonably addressed through clear descriptions and standard datasets, though explicit code release is not mentioned.

Guillory & Bilmes, 2009: Label selection on graphs. Cesa-Bianchi et al., 2010: Active learning on trees and graphs.

**Methods And Evaluation Criteria:**

The proposed methodology leverages a reduction of the graph label selection problem to a **flow-based formulation** inspired by the densest subgraph problem. The authors introduce a *flow gadget*, where labeling vertices translates into adding edges to increase the max-flow, enabling a greedy selection process. Though the original objective $\Psi(L)$ is neither submodular nor supermodular, the derived surrogate measure (flow increase) is submodular, allowing established submodular optimization guarantees. They complement this with a binary search approach to efficiently pinpoint feasible threshold values. Technical assumptions (like polynomially bounded edge weights) are clearly stated and justified, with full details provided in appendices.

The evaluation methods—worst-case approximation analysis and empirical $\Psi(L)$ values—are appropriate for the theoretical nature of the paper. Empirically, the authors demonstrate improvements over baseline methods through controlled experiments on both synthetic and real-world graphs. While the evaluation does not directly address prediction accuracy, the chosen proxy ($\Psi(L)$) suitably reflects the intended smoothness goal. Overall, the methods and evaluation criteria are thoughtfully designed, effectively validating the proposed approach.

**Other Comments Or Suggestions:**

N/A

**Other Strengths And Weaknesses:**

**Strengths:**

- Theoretical novelty: providing approximation guarantees for general graphs.
- Creative algorithmic insights: clever use of max-flow reduction and submodular optimization framework.
- Well-presented theoretical results with careful, rigorous reasoning.
- Useful extension to vertex importance scenarios.

**Weaknesses:**

- *Scalability:* A notable weakness is that the proposed algorithm is **computationally intensive**, with a runtime that is super-linear in the number of vertices. The authors admit it doesn’t scale to very large graphs. In an era where graphs can have millions of nodes, an algorithm that likely struggles beyond a few thousand nodes has limited immediate application. This is somewhat inherent given the complexity of the problem, but it does mean the practical impact is curtailed until further optimizations are found.
- *Resource Augmentation Requirement:* The approximation guarantee is achieved by using up to $O(k \log n)$ labels instead of $k$. From a theoretical perspective this is fine (and indeed common in approximation algorithms), but from a practical standpoint, it means the method might need substantially more labeled data than one’s budget in order to guarantee optimal results. In the experiments, they ran it with exactly $k$ labels and it still outperformed others, but there’s no guarantee in that regime. In scenarios where label budget is strict, one might wonder how suboptimal the greedy picks could be if cut off at $k$. This gap between theory (with augmentation) and practice (fixed budget) is worth noting as a weakness, though the empirical evidence suggests the algorithm still performs well without augmentation.
- *Focus on Batch Selection:* The paper addresses the *offline* (batch) selection problem. It does not consider the fully sequential active learning setting where one can query a label, update the model, and then choose the next query. The batch selection is an important scenario on its own (and often needed for parallel labeling), but it’s inherently a restricted version of active learning. The contribution is still valuable, but it doesn’t solve the entire active learning problem on graphs – only the one-shot selection version. Future work might need to explore how these results extend (or not) to an interactive setting.
- *Minor Clarity Issues:* While generally well-written, some parts of the proof sketches and the description of the approach are dense. For instance, the flow gadget construction might be hard to follow for readers not versed in network flow. The paper relies on the appendix for full clarity. Additionally, some terminology like “constant slack” (used to describe the prior tree algorithm’s approximation) or “resource augmentation” could have been defined in a more reader-friendly way early on. These are relatively minor weaknesses in exposition.
- *Experiment Limitations:* The experiments, as mentioned, are on small graphs and a limited number of datasets. The paper might have benefited from at least one medium-scale experiment (if feasible) or some discussion on how the algorithm behaves as graph size grows (e.g., a plot of runtime or objective vs. $n$). Reproducibility could also be enhanced by providing pseudocode (the algorithm is described in prose but pseudocode could help implementers) – though perhaps it’s included in the appendix or could be derived from the description.

**Questions For Authors:**

N/A

**Relation To Broader Scientific Literature:**

The paper positions itself clearly within existing literature, referencing both foundational works and recent advancements in relevant fields, specifically:

- **Graph-based Semi-supervised Learning:**
  (Blum & Chawla, 2001; Zhu et al., 2003; Belkin et al., 2004; Bengio et al., 2006).
- **Active Learning on Graphs:**
  (Guillory & Bilmes, 2009; Cesa-Bianchi et al., 2010; Dasarathy et al., 2015)
- **General and Deep Active Learning:**
  (Settles, 2009; Ren et al., 2021; Mac Aodha et al., 2014; Kushnir & Venturi, 2020)
- **Algorithmic Graph Theory (densest subgraph):**
  (Goldberg, 1984; Boob et al., 2020; Chekuri et al., 2022)

Overall, the paper effectively builds upon previous theoretical frameworks (e.g., Guillory & Bilmes, 2009; Cesa-Bianchi et al., 2010) and clearly outlines how it extends prior limitations, such as handling general graphs rather than special cases (trees). While the paper does not deeply explore sequential active learning, this omission is justified given its scope.

### Standard Citations:

- Blum, A., & Chawla, S. (2001). *Learning from labeled and unlabeled data using graph mincuts.* ICML.
- Zhu, X., Ghahramani, Z., & Lafferty, J. (2003). *Semi-supervised learning using Gaussian fields and harmonic functions.* ICML.
- Belkin, M., Matveeva, I., & Niyogi, P. (2004). *Regularization and semi-supervised learning on large graphs.* COLT.
- Guillory, A., & Bilmes, J. A. (2009). *Label selection on graphs.* NeurIPS.
- Cesa-Bianchi, N., Gentile, C., Vitale, F., & Zappella, G. (2010). *Active learning on trees and graphs.* COLT.
- Dasarathy, G., Nowak, R., & Zhu, X. (2015). *S2: An efficient graph-based active learning algorithm.* COLT.
- Settles, B. (2009). *Active learning literature survey.* University of Wisconsin-Madison.
- Ren, P., et al. (2021). *A survey of deep active learning.* ACM Computing Surveys.
- Mac Aodha, O., Campbell, N. D., Kautz, J., & Brostow, G. J. (2014). *Hierarchical subquery evaluation for active learning on a graph.* CVPR.
- Kushnir, D., & Venturi, L. (2020). *Diffusion-based deep active learning.* arXiv preprint arXiv:2003.10339.
- Goldberg, A. V. (1984). *Finding a maximum density subgraph.* Technical report, UC Berkeley.
- Boob, D., et al. (2020). *Flowless: Extracting densest subgraphs without flow computations.* WWW.

Overall, the literature review is thorough and clearly situates the paper’s contribution within the broader field.

**Theoretical Claims:**

The theoretical claims are significant and appear correct based on the provided material. The main results include:

- **Approximation Algorithm (Theorem 1.1):** By allowing a factor-$O(\log n)$ more labels than the budget $k$, the algorithm achieves an objective equal to the optimal solution for exactly $k$ labels. The authors provide a plausible proof sketch (detailed proofs in Appendix B), relying on standard submodularity and flow techniques. The assumptions (e.g., polynomially bounded integral weights) are clearly stated and reasonable.

- **Hardness Result (Theorem 1.2):** The authors show it is NP-hard to distinguish between cases where the optimal value is at most 2 or at least 3, implying constant-factor hardness of approximation. This reduction from known NP-hard problems on 3-regular graphs seems correct and convincing, clearly establishing the problem’s theoretical difficulty.

- **Generalization to Weighted Vertex Importance:** The paper extends the approximation results to scenarios with vertex importance weights, offering a generalization that appears logically sound and consistent, though I did not verify proofs exhaustively.

Overall, the theoretical contributions are rigorous, clearly presented, and supported by well-known techniques. While the runtime complexity ($(|V|+|E|)^{1+o(1)}$ per iteration) is slightly super-linear and somewhat limits practical scalability, it does not undermine the theoretical validity of the results presented.

---

> ### Author Rebuttal · Authors · 2025-03-30
>
> Dear reviewer qHp3,
>
> Thank you for your thoughtful review, we will address your comments in the final version. Here we will focus on responding to the points you raised.
>
> 1) Scalability: We agree that the scalability, together with removing the resource augmentation requirement, is the main research question left open by our work. Given the progress on linear time algorithms for the related densest subgraph problem, we hope that further work can remove this deficit. This might come at the cost of further relaxing the theoretical guarantees.
>
> 2) Resource Augmentation Requirement: We agree with your assessment that a proper approximation algorithm that does not expand the budget would be interesting from a theoretical and practical point of view. A theoretical reason for the observed practical performance of our algorithm in this regime is that a resource augmented algorithm is also competitive for many values of $k$. This phenomenon is observed in a classic text by Tim Roughgarden and called loosely competitive (https://arxiv.org/abs/2007.1323).
>
> 3) Experiment Limitations: Due to the large number of maximum flow computations, we were unable to scale up our experiments to larger graphs. In our implementation (see supplementary material) we already parallelize the implementation of the algorithm to enable our current experiments in a reasonable timescale. We hope that future work motivated by these preliminary experiments will significantly reduce the time complexity.
>
> 4) Focus on Batch Selection: We focus on the batch selection problem as previous work in the area. We agree that an appropriate model of adaptivity could serve as a starting point for further fruitful research. We will leave it as an open direction for future work in the final version
>
> 5) Minor Clarity Issues: We will go over the writeup again to further improve the exposition and clarity of the proofs.
>
> 6) Experiment Limitations: Using maximum flow as a subroutine, and calling it at least as often as there are vertices, unfortunately means that this algorithm does not scale to large graphs as is. We already parallelized part of the implementation (see supplementary material) to enable the current experiments. As discussed in the first point, improving the scalability is a major open problem, now that a polynomial time solution has been found.

---

### Decision · Program_Chairs · 2025-05-01

**Decision:**

Accept (poster)

**Comment:**

All reviewers agree that the paper is interesting and worth publishing at ICML. I agree with the assessment as well.